

# Sensitivity of forecast skill to the parameterisation of moist convection in a limited-area ensemble forecast system

Matteo Vasconi[1,2], Andrea Montani[2], and Tiziana Paccagnella[2]

[1]Department of Physics and Astronomy, University of Bologna, Italy
[2]Arpae-SIMC, HydroMeteoClimate Service of the Emilia-Romagna Region, Bologna, Italy

**Correspondence:** Matteo Vasconi (matteo.vasconi@unimore.it)

**Abstract.** The parameterisation of convection in limited-area models is an important source of uncertainty as regards the spatio-temporal forecast of precipitation. As for the limited-area model COSMO, hitherto, only the Tiedtke convection scheme was available for the operational runs of the model in convection-parameterised mode. In addition to this the Bechtold scheme, implemented in ECMWF global model, has recently been adapted for COSMO applications. The development and implementation of ensemble systems in which different convection schemes are used, provides an opportunity to upgrade state-of-the-art probabilistic systems at the convection-parameterised scale. The sensitivity of the COSMO model forecast skill to the use of either the Tietdke or the Bechtold schemes is assessed by performing different sets of experiments.

The performance of COSMO model run with the different schemes is investigated in ensemble mode with particular attention to the types of forecast errors (e.g. location, timing, intensity) provided by the different convection schemes in terms of total precipitation.

A 10-member ensemble has been run for approximately 2 months with the Bechtold scheme, using the same initial and boundary conditions as members 1-10 of the operational COSMO-LEPS ensemble system (which has 20 members, all run with the Tiedtke scheme). The performance of these members is assessed and compared to that of the system made of members 1-10 of COSMO-LEPS in terms of total precipitation prediction.

Finally, the performance of an experimental 20-member ensemble system (which has 10 members run with the Bechtold plus 10 members run with the Tiedtke scheme) is compared to that of operational COSMO-LEPS over the 2-month period. The new system turned out to have higher skill in terms of precipitation forecast with respect to COSMO-LEPS over the period. In this approach the use of the Bechtold scheme is proposed as a perturbation for the COSMO-LEPS ensemble, relatively to how uncertainties in the model representation of the cumulus convection can be described and quantified.

## 1 Introduction

Numerical Weather Prediction (NWP) models have been developed over the last 50 years in order to quantitatively predict the future states of the atmosphere using the current weather conditions. Despite the constant increase in horizontal and vertical resolutions of these models, the accurate forecast of high-impact weather still remains difficult beyond day 2 and, sometimes, also for shorter ranges (Mullen and Buizza, 2001; Tibaldi et al., 2006). Several factors contribute to forecast failures and can

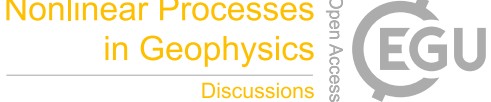



be usually related to shortcomings in the definition of the initial conditions of the integrations, to model errors of different types and, last but not least, to the intrinsic low predictability of the physical phenomena under investigation. In order to tackle the chaotic behaviour of the atmosphere and to support forecasters in the management of alert procedures for events with little deterministic predictability, ensemble forecasting was introduced at the beginning of the nineties. Instead of running just

one forecast with an unknown error, an ensemble of slightly different forecasts are run, in order to integrate the deterministic forecast with an estimate of the "forecast of forecast skill" (Tennekes, 1986). The estimation of uncertainty is even more crucial when a high spatio-temporal detail is required as in the case of precipitation. One of the most important source of errors is the approximate model formulation of some physical processes which have typical horizontal dimensions too small to be explicitly resolved. In fact the only way to represent the overall effects of these sub-grid physical processes in NWP models is

by means of parameterisations. From an operational point of view, one of the most interesting parameterisation is that relative to moist convection (cumulus convection) because it regards the spatio-temporal forecast of precipitation. Convective processes typically operate on horizontal scales which are much smaller than those resolved by large-scale and mesoscale NWP models, even though current and next generation limited area versions of models can use horizontal resolutions of O(1-3 km), and therefore can resolve at least deep convection with reasonable accuracy.

The common point of all cumulus parameterisations is that they aim at diagnosing the presence of larger-scale conditions that would support the development of convective activity and at calculating the collective effects of an ensemble of convective clouds in a model column as a function of grid-scale variables. In particular, most parameterisations are designed to drive the model atmosphere towards a convectively adjusted state when they activate, by removing convective instability and producing subgrid-scale convective precipitation in unsaturated model grids. Many methods have been implemented in NWP models

over the last 40 years to represent moist convection processes. The different approaches mainly differ in terms of cloud model, closure assumptions and computational efficiency. The progress in this aspect of atmospheric modeling has been especially slow over the past decades (Randall et al., 2003) because, in addition to the basic question of how to pose the problem, there are several uncertainties in modeling these processes, as reviewed by Arakawa (2004). This is a signature of both the complexity of the physical processes involved and the uncertainty linked to the use of a particular scheme instead of another.

As for COSMO (COnsortium for Small-scale MOdelling, http://www.cosmo-model.org) model applications, either the Tietdke scheme (Tiedtke, 1989), which has been operationally used so far, or the Bechtold scheme (Bechtold et al., 2001; 2014), initially implemented for the ECMWF (European Centre for Medium-range Weather Forecast) global model and recently adapted to COSMO model, are now available.

Both methods use a mass-flux approach to represent moist convection in numerical models. The mass flux approach is a gen-

eral and quite powerful method to tackle (eddy) transport problems in fluid mechanics, in particular for convective overturning where most of the transport is done by the "large" eddy draughts that carry heat and mass upward and downward over relatively large distances (Bechtold, 2017). The feedback of subgrid-scale vertical fluxes of mass, heat, moisture and momentum in up- and downdraughts is then calculated by using a simple bulk cloud model (Fig. 1). The main differences between the two schemes are in the closure assumptions. Neverthless both methods discriminate three types of convection: deep, shallow and

midlevel convection, which are treated by different closure hypotheses. In particular, in case of deep convection, an equilibrium



type of closure is applied in the Tiedtke scheme by imposing a moisture balance for the subcloud layer such that the vertically integrated specific humidity is maintained in the presence of grid-scale, turbulent and convective transports (Kuo-type closure; Kuo et al., 1980). On the other hand, an equilibrium between the large-scale and boundary-layer forcing (generating convective available potential energy) and convection (reducing the CAPE, Convective Available Potential Energy) is assumed in the Bechtold formulation (Bechtold, 2017).

In addition to this, it is worth pointing out is that while the Tiedtke scheme treats shallow non-precipitation convection only, the Bechtold scheme allows "shallow convection" to produce precipitation.

The aim of this work is to assess the sensitivity of the COSMO model forecast skill to the use of these two different parameterisation of moist convection in ensemble mode, by assessing the ability of the system to predict precipitation events. As already pointed out the parameterisation of convection in limited-area models is an important source of uncertainty as regards the spatio-temporal forecast of precipitation. Therefore the development and implementation of multi-physics ensemble systems where two different schemes can be used by the ensemble members, provides an opportunity to upgrade state-of-the-art probabilistic systems at the convection-parameterised scale and, in particular, COSMO-LEPS (COnsortium for Small-Scale MOdelling Limited-area Ensemble Prediction System), the operational ensemble system of the consortium, which uses only the Tiedke scheme (Montani et al., 2011). Ensembles using multiple model formulations can provide better estimate of uncertainty in the model physics, facilitating the reduction of forecast errors, helping to take into account all the possible future states of the atmosphere and providing a more reliable estimate of the day-to-day forecast skill.

The paper is organised as follows: section 2 presents the model description and the set-up of the dexperiments, while section 3 reports the main results. Finally, conclusions are drawn in section 4.

## 2 Model description and experiments

### 2.1 Model system

The COSMO-Model is a non-hydrostatic limited-area atmospheric prediction model, based on the primitive thermo-hydrodynamical equations describing compressible flow in a moist atmosphere, with a variety of physical processes taken into account by parameterisation schemes (Doms et al., 2015). As far as operational implementations are concerned, the COSMO-LEPS was the first mesoscale ensemble application running on a daily basis in Europe. This system, initially developed and implemented by the HydroMeteoClimate Regional Service of Emilia-Romagna, in Bologna, Italy (Arpae-SIMC), has been running at ECMWF since November 2002 (Montani et al., 2003a). Nowadays, COSMO-LEPS is based on 20 integrations of the non-hydrostatic mesoscale model COSMO, formerly known as the Lokal Modell (Steppeler et al., 2003). The methodology adopted in this system aims at combining the advantages of the probabilistic approach by global ensemble systems with the high-resolution details gained in the mesoscale integrations. In the construction of COSMO-LEPS, an algorithm selects a number of members (referred to as Representative Members, RMs) from ECMWF-ENS global ensemble system (Marsigli et al., 2001). After the "ensemble-size reduction", the selected RMs are used to provide both initial and boundary conditions to the integrations with



the COSMO model, which is run once for each RM. For more details on the methodology the reader is referred to Montani et al. (2011).

### 2.1.1 Description of the experiments

Some experiments have been performed, in order to evaluate the COSMO model performance in ensemble mode when it is run

either with the Tiedtke or the Bechtold scheme, so as to assess overall abilities and shortcomings of the system (Vasconi, 2017). Firstly, we have built a test suite to run a 10-member ensemble with the Bechtold scheme (referred to as Cleps-10B), which uses the same initial and boundary conditions as members 1-10 of the operational COSMO-LEPS (which has 20 members, all run with the Tiedtke scheme). This suite has been run from 28$^{th}$ March to 31$^{th}$ May 2017 with an integration domain covering Central-Southern Europe and Italy (shown in Fig. 2), at the horizontal resolution of about 7 km and 40 vertical

layers, and with a 132-hours forecast range, always starting at 00 UTC. In particular, the sensitivity of the ensemble system to the different parameterisation schemes has been assessed by comparing the performance of Cleps-10B to that of Cleps-10T, which is the 10-member ensemble provided by members 1-10 of COSMO-LEPS, the operational ensemble system of the COSMO consortium, over the verification period. A further step in the study of COSMO ensemble system sensitivity to different formulation of moist convection is the implementation of a new probabilistic system, hereafter Cleps20bt, in which

a multi-physics approach in the model representation of the cumulus convection is followed. This system is generated by adding the members of Cleps-10B to members 11-20 of COSMO-LEPS. Therefore, Cleps20bt has 10 members run with the Bechtold scheme plus 10 members run with the Tiedtke scheme and no duplication of initial and boundary conditions. The basic idea of the Cleps20bt implementation is that certain closure parameters used in model formulation (as for the moist convective processes) may be based on approximate physical knowledge. As a consequence their values may be somewhat

arbitrary, or they may have been tuned to give optimal results for test cases that are not necessarily representative of more general applications and/or for applications at high resolution. A summary of the ensembles features is presented in Table 1.

### 2.2 Methodology of verification

The performance of the ensemble systems was analysed by considering the probabilistic prediction of 6-h cumulated precipitation exceeding a number of thresholds for forecast up to 132 hours over the 2-month period.

Since precipitation has a high-spatial variability, a high-density network, made of about 1000 stations over Northern Italy (Fig. 3), has been adopted in order to assess the predictive skill of the ensemble systems. For the comparison of the model forecasts against station reports the grid point closest to the observation one is selected. In particular the performance of the different ensemble systems of Table 2 is examined for six different 6-h cumulated precipitation thresholds: 1, 5, 10, 15, 25, 50 mm/6-h. Several thousands of events were reported for the first two thresholds, and several hundreds for the 15 mm/6-h

threshold. On the other hand it is immediately worth pointing out that, when considering the highest thresholds (25, 50 mm/6-h), a low number of occurrences, even below 10 for the 50 mm/6-h, was found over the verification period. As a consequence this does not allow any solid statistical conclusion on the effective performance of the system for these events over the period.




For each forecast range, the model performance has been evaluated by computing the following "traditional" probabilistic scores (Wilks, 1995): the Brier Skill Score (BSS), the Ranked Probability Skill Score (RPSS), and the Percentage of Outliers (Buizza, 1997). A summary table of the verification features is reported in Table 2.

## 3   Results

### 3.1   Comparison of 10-member ensemble system run with different schemes

The BSS (Brier Skill Score) for the Cleps-10T and Cleps-10B is presented in Fig. 4. A 24-h running mean is here applied to "smooth" the diurnal cycle in model performance, improving the readability of the plot. This score tries to represent a quantitative estimate of the added value detectable in precipitation prediction by using the model forecast rather than a reference one (in this case, climatology of the observed sample over the verification period). The attention has been focused on two thresholds (1 mm/6-h and 15 mm/6-h), which have a quite large number of occurences (higher than 1000 for the former, some hundreds for the latter) over the verification period.

It is worth noticing that the BSS shows clearly the loss of predictability with increasing forecast range for both systems. The model forecast has added value with respect to the reference climatology up to +120 hours. However the plot shows a different skill of the 2 systems when different thresholds and forecast ranges are considered. Over the verification period, Cleps-10T performs generally better than Cleps-10B for the lower threshold (1mm/6-h), while the opposite is true in high precipitation rates prediction for forecast ranges from 3 days onwards. In other words, the ensemble systems seem to describe different types of forecast errors, possibly related to the different convection schemes (Vasconi, 2017).

In addition to this, the RPSS (Ranked Probability Skill Score) of this system has been computed for different forecast ranges and compared to that of COSMO-LEPS during the same period. The plot in Fig. 5 shows a better performance of Cleps-10T for the forecast ranges up to +48 hours.

These results can be seen consistent with the theory according to which the ensemble systems which are run using either convection schemes can describe a larger variety of uncertainty and errors in precipitation prediction.

Finally, the skill of the two systems has been assessed in terms of Percentage of Outliers (that is the cases in which observed rainfall value is not inside the ranges of possible values predicted by the ensemble members, Fig. 6). Firstly it is worth pointing out that the total percentage of outliers (left panel) for both systems tends to decrease with increasing forecast range because of the increasing spread with time between the ensemble members. A better performance of Cleps-10T, which has a lower number of outliers than Cleps-10B, can be noticed, in particular for the earlier forecast ranges. The right panel of Fig. 6 represents respectively the fraction of points in which observations lie above/below the range of predicted values by the ensemble system. A large amount of outliers below the minimum forecast value, indicative of an overestimation of minima of precipitation amount by Cleps-10B runs, can be seen. In particular the percentage of outliers lying below the minimum predicted values is higher for Cleps-10B than for Cleps-10T for all the forecast ranges studied. This seems to indicate that members with the Bechtold scheme tend to produce some light prepitation also when it is not observed. On the other hand, the fraction of analysis point above the maximum tends to be similar or slightly lower for Cleps-10B. This excessive drizzle effect could be due to the



shallow convection treatment adopted by the Bechtold scheme. This scheme in fact allows "shallow convection" to produce precipitation, whereas the Tiedtke scheme does not. It is possible that further tuning of the Bechtold scheme, when adopted at high resolution, is necessary to address this "drizzle" issue.

### 3.2 Performance of Cleps20bt and comparison with that of COSMO-LEPS

A quantitative evaluation of Cleps20bt skill in terms of precipitation forecast over the the same period is then presented. The basic idea of this study is that ensemble systems which are run using either convection schemes can describe a larger variety of uncertainty and errors in precipitation prediction (Vasconi, 2017). Thus the implementation of ensemble systems in which the two schemes are "mixed" seems to be a reasonable issue to deal with uncertainties due to the ambiguity linked to the use of a scheme or the other. It is worth pointing out that the implementation of this experimental system is consistent only because the

average skill of the model when it is run in ensemble mode with the Bechtold scheme turned out to be roughly indistinguishable, from a statistical point of view, from that provided by running the model with the Tiedtke scheme, as shown in the previous Section. In fact, in a well-constructed ensemble, the skill of each individual member, averaged over a large number of events, should be approximately identical not to introduced biases and/or systematic errors in the ensemble members distribution.

The forecast skill in terms of precipitation of Cleps20bt is then assessed and compared to that of COSMO-LEPS. The main

results of this study are presented in the following plots. In Fig. 7 BSS (Brier Skill Score) is presented for different forecast ranges by considering several thresholds. In particular the focus is on the same threshold as for the 10-member case, for which a relative large number of events has been reported (1 mm/6-h and 15 mm/6-h). In order to provide an overall description of the model system performance for the different precipitation thresholds, the values reported in the plot are obtained, once again, by computing the running mean of the 6-h precipitation forecast skill over 24 hours. The plot shows that Cleps20bt has higher

values of BSS than COSMO-LEPS for the thresholds reported, especially for forecast ranges from 42 hours onwards (blue and red lines respectively).

In addition to this, the RPSS (Ranked Probability Skill Score) of this system has been computed for different forecast ranges and compared to that of COSMO-LEPS during the same period. The comparison between the 24-h running mean of RPSS for the two systems is presented in Fig. 8. Also in this case a better performance of Cleps20bt than that of COSMO-LEPS is

evident for forecast ranges from 2 days onwards: for example RPSS in the forecast range +60-66 hours is about 5% higher in Cleps20bt than in COSMO-LEPS; it is about 10% higher in the new system for +90-96 h, +96-102 h ranges.

A similar behaviour can be detectable also in other scores (Brier Score and ROC Area), which are not presented in this paper.

Finally the performance of the systems is evaluated in terms of the percentage of outliers (left panel in Fig. 9). In addition to this, similarly to the 10-member ensembles case, the percentage of outliers are discriminated between the fractions of points in

which observed values lay outside the forecast range over the full verification period (right panel in Fig. 9). The percentage of outliers is reduced in Cleps20bt over most of the forecast ranges with respect to COSMO-LEPS, especially from 3 days (+72 hours) onwards.

The right panel in Fig. 9 shows that the total percentage of outliers is reduced in Cleps20bt as a consequence of a decrease in the number of points wherethe total precipitation maxima are underestimated compared to COSMO-LEPS. In fact the fraction





of observations found above the maximum forecast value is lower in Cleps20bt than in COSMO-LEPS, for most of forecast ranges, especially in the medium range (from +72 hours onwards). This is a quite encouraging result because Cleps-20bt turns out to perform better than the operational COSMO-LEPS in forecasting the possible peaks in cumulated precipitation over the 2-month period. It is worth underlining that the probabilistic forecast of these values is one of the most important issue of

5   operational systems, because it regards the correct prediction of heavy rainfall events, which may have a high impact on the society.

This result, together with those presented in this section, substantially agrees with the idea that, by adding a physical perturbation to the system (like what we have done in this work using an ensemble system in which two different moist convective schemes are used), we can obtain a more appropriate description of the phase-space of all possible future atmospheric states

10   which are compatible with the uncertain model formulation of the moist convection sub-grid processes. Thus, according to this experimentation, the generation of a multi-physics ensemble system provides a positive impact on the forecast capability at high resolution. This is especially true in early-medium range, when model errors start playing an important role and it is crucial for an ensemble system to provide an accurate description of the different sources of forecast deficiency (Vasconi, 2017).





## 4   Conclusions

The impact of the use of two moist convection schemes (the Tiedtke and Bechtold schemes) has been studied in ensemble mode. Firstly a 10-member ensemble with the Becthtold scheme (Cleps-10B), which uses the same initial and boundary conditions as members 1-10 of the operational COSMO-LEPS, has been run has been run for approximately 2 months. The performance

of these members has been assessed and compared again to that of Cleps-10T, the 10-member ensemble made of members 1-10 of COSMO-LEPS; in particular the spread/skill relation of the two 10-member ensemble in terms of total precipitation is evaluated. Verification has been performed for precipitation events occurred over Northern Italy (using the forecast at the gridpoints nearest to about 1000 stations) from $28^{th}$ March to $31^{th}$ May 2017. The average skill of the Cleps-10B runs turned out to be substantially indistinguishable, from a statistical point of view, from that provided by the Cleps-10T ones. However a

deeper analysis suggests that the two ensemble systems are characterised by different types of forecast errors. Therefore a new 20-member ensemble system (Cleps20bt, which has 10 members run with Bechtold plus 10 members run with Tiedtke and no duplication of boundary conditions) has been implemented. In this system the Bechtold scheme is used as a perturbation for the COSMO-LEPS ensemble, so as to provide a quantitative description of uncertainties linked to the model representation of the cumulus convection. Cleps20bt has been shown to have higher skill than COSMO-LEPS over the verification period. In

addition to this, the comparison of the Percentage of Outliers in the two systems shows a reduction in the fraction of observed points lying outside the maximum or minimum forecast value in Cleps20bt. These results suggest that the use of a probabilistic system in which a multiple moist convection formulation is used, provides the opportunity to have a more comprehensive description of the uncertainties in total precipitation forecast linked to the sub-grid cumulus representation.

*Competing interests.*   The authors declare that they have no conflict of interest.

*Acknowledgements.*   The authors are grateful to Prof. Silvana Di Sabatino and Dr. Chiara Marsigli for useful discussions and to Dr. Davide Cesari for technical assistance.



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



| Acronym | Ensemble size | Convection scheme | ICs-BCs |
|---|---|---|---|
| COSMO-LEPS | 20 | Tiedtke | from ECMWF-ENS |
| Cleps-10B | 10 | Bechtold | the same as members 1-10 of COSMO-LEPS |
| Cleps-10T | 10 | Tiedtke | the same as members 1-10 of COSMO-LEPS |
| Cleps-20bt | 20 | members 1-10: Bechtold, members 11-20: Tiedtke | the same as COSMO-LEPS |

**Table 1.** Main features of the ensemble systems of Section 2

| Verification features | |
|---|---|
| variable: | 6-h cumulated precipitation (00-06, 06-12,..UTC); |
| Period: | from 28[th] March to 31[th] May 2017 (about 60 days); |
| region: | Northern Italy; |
| method: | nearest grid-point; no-weighted fcst; |
| obs: | non-GTS network, no obs error; |
| fcst ranges: | 0-6 h, 6-12 h,..., 126-132 h; |
| thresholds: | 1, 5, 10, 15, 25, 50 mm/6 h; |
| systems: | Cleps-10B vs Cleps-10T, Cleps20bt vs COSMO-LEPS; |
| scores: | BSS, RPSS, Percentage of Outliers. |

**Table 2.** Main features of the verification configuration for the ensembles

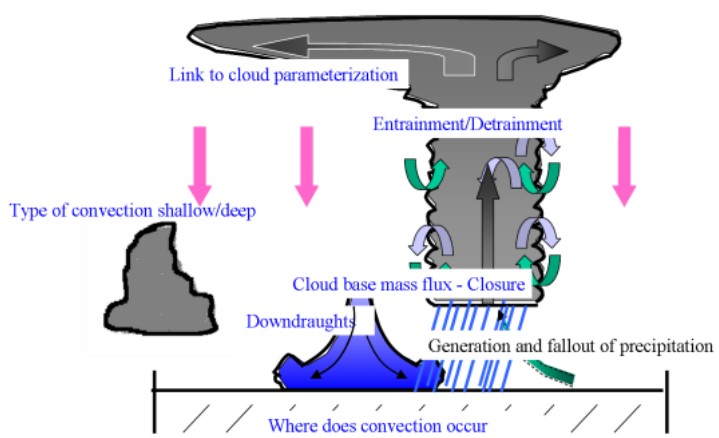

**Figure 1.** Schematic of a bulk convection scheme with a shallow and deep entraining/detraining cloudy ascending plume, and downdraught region. Further represented features are trigger of convection, environmental subsidence, microphysics and precipitation, and detrainment of cloud mass in anvils (Bechtold, 2017).

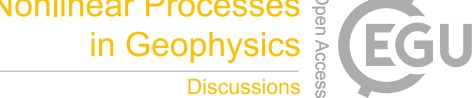



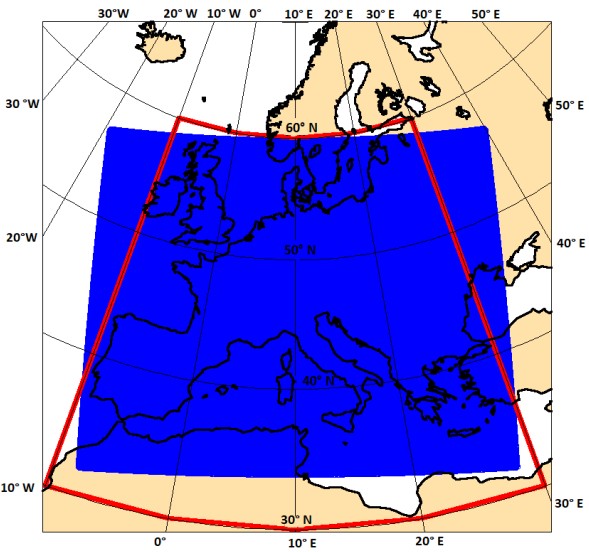

**Figure 2.** COSMO-LEPS integration domain (blue area) and clustering area (inside the red line).

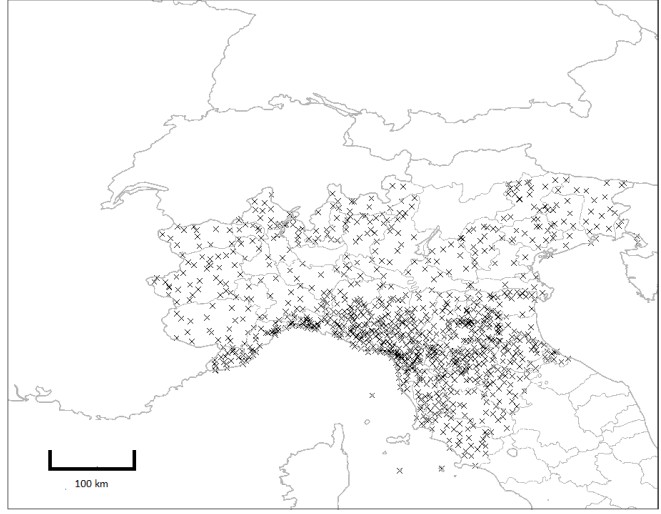

**Figure 3.** Observation network used for verification.





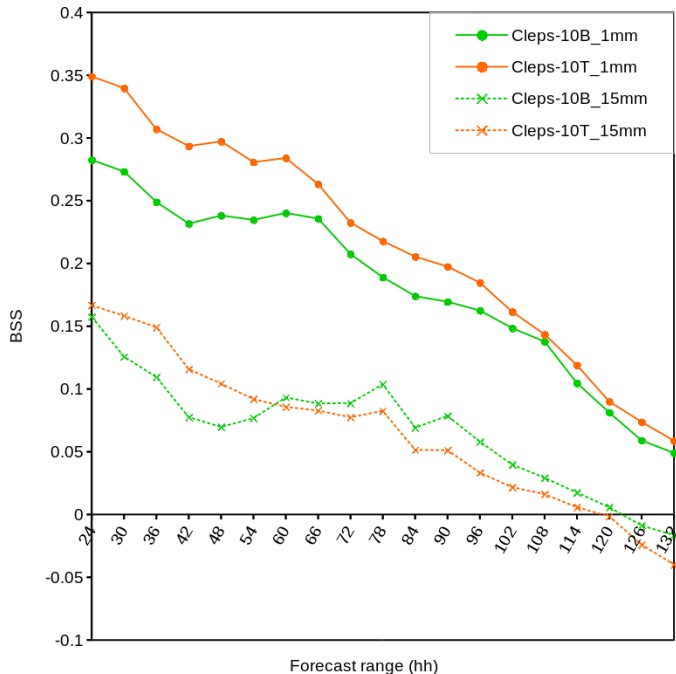

**Figure 4.** 24-h running mean of BSS in Cleps-10T and Cleps-10B (orange and green line, respectively) for 1 mm/6-h and 15 mm/6-h (solid and dashed line, respectively) thresholds.

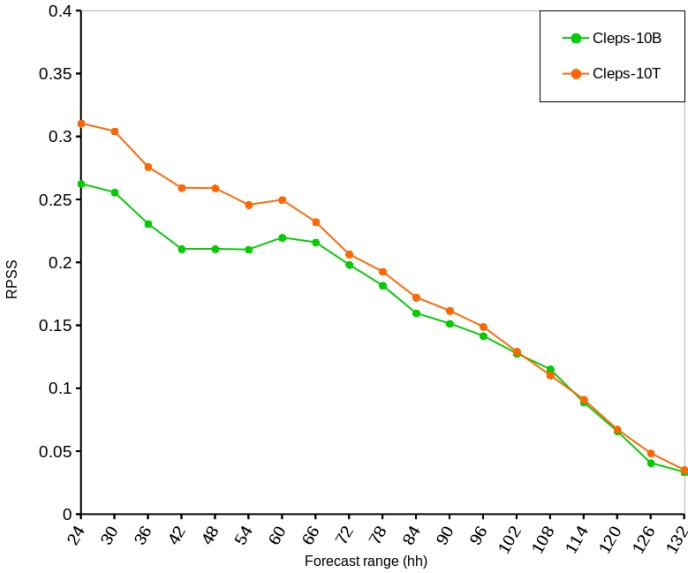

**Figure 5.** 24-h running mean of RPSS in Cleps-10T (orange line) and Cleps-10B (green line).



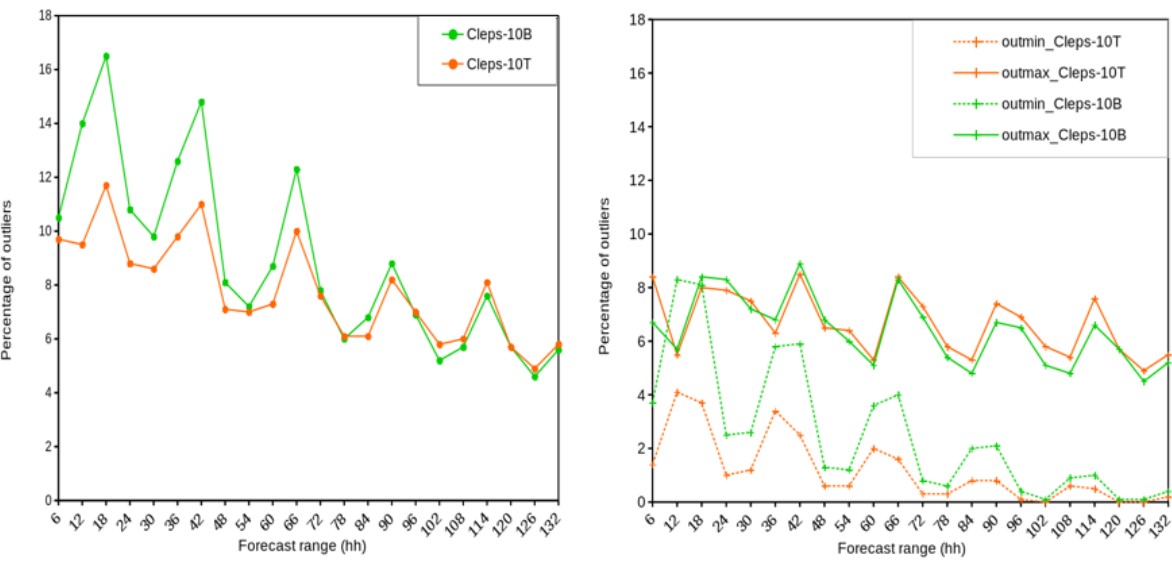

**Figure 6.** Left panel: Percentage of outliers for different forecast ranges in Cleps-10T and Cleps-10B (orange and green line, respectively). Right panel: Percentage of outliers above/below maximum/minimum predicted values

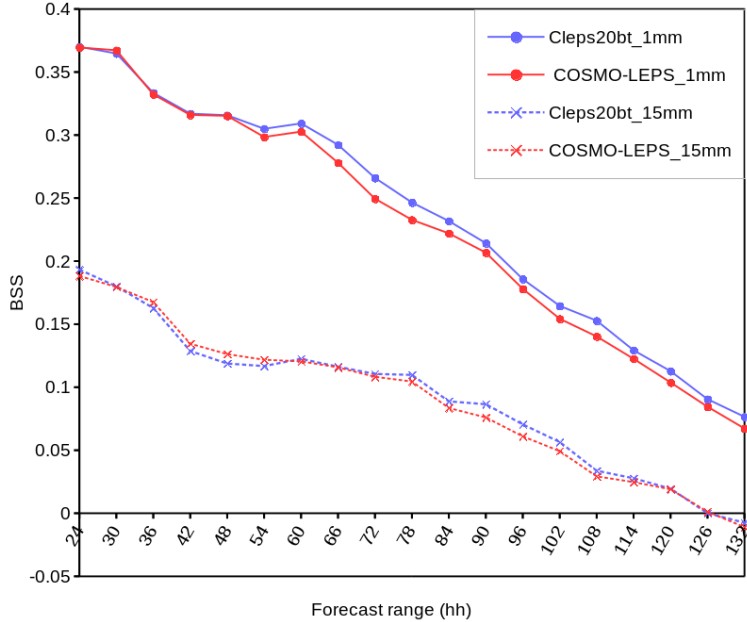

**Figure 7.** 24-h running mean of BSS values for 6-h accumulated precipitation exceeding 1 mm and 15 mm (solid and dashed line, respectively) for different forecast ranges in COSMO-LEPS (red line) and Cleps20bt (blue line).





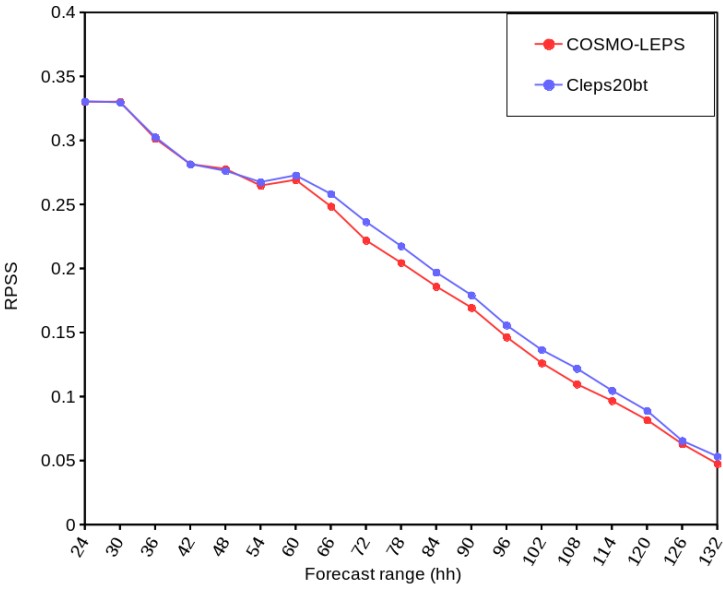

**Figure 8.** 24-h running mean of RPSS values for 6-h accumulated precipitation for different forecast ranges in COSMO-LEPS (red line) and Cleps20bt (blue line).

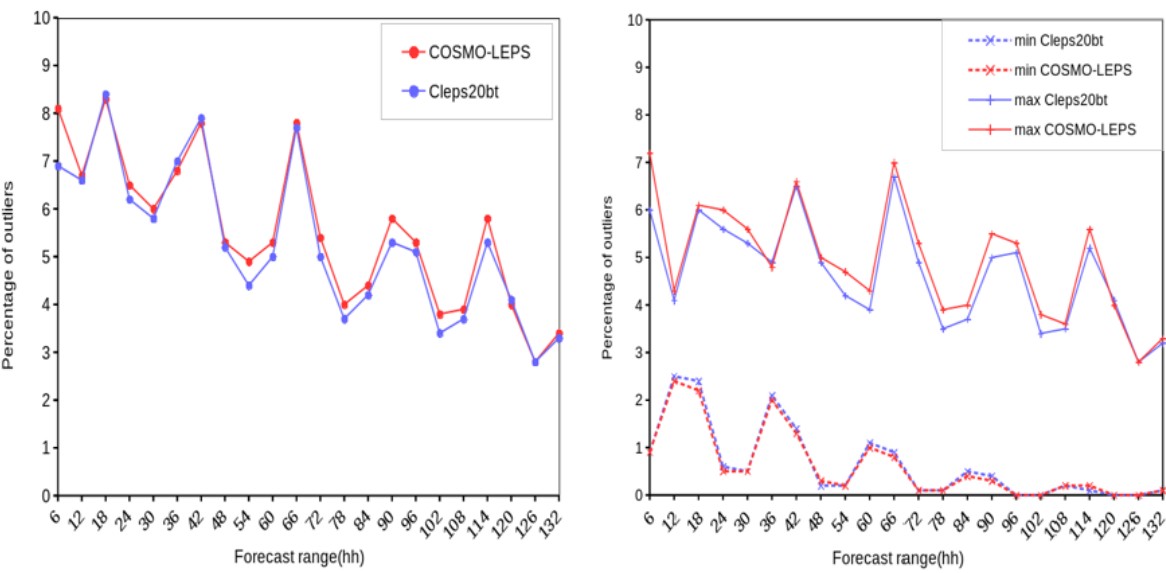

**Figure 9.** Left panel: Percentage of outliers for different forecast ranges in COSMO-LEPS (red line) and Cleps20bt (blue line). Right panel: Percentage of outliers above/below maximum/minimum predicted values.