# Peer review of "Sensitivity of forecast skill to the parameterisation of moist convection in a limited-area ensemble forecast system"

_Nonlinear Processes in Geophysics, 2018_

## Referee Comment (RC1) · Anonymous Referee #1 · 27 Mar 2018

The manuscript by Vasconi et al. is a short and useful contribution on the skill of limited area ensembles in predicting summertime precipitation and its dependence on the formulation of subgrid convective transport.

I only have some minor comments and corrections:

Title: change "forecast skill" in "precipitation forecast skill"

-Abstract line 3: I suggest to change "Bechtold scheme,.. in ECMWF model" in something like "an evolved" or "revised version of the Tiedtke scheme, referred to as Bechtold scheme as implemented in the ECMWF global model"

[Figure]

-the appropriate references are Bechtold et al. (2008, QJRMS) and Bechtold et al. (2014, JAS). Please delete reference to Bechtold et al. (2001) as this article was on a completely different scheme

-the main shortcoming of the "Bechtold" scheme in COSMO is as you riightly discuss the overestimation of drizzle over land - while it shold in principle provide a much improved climatology overall including 2m T. Maybe it is worthwhile to add a Figure with a Pdf of rainfall, from Tiedtke, Bechtold and Obs. Is there anything to say on other surface variables, maybe in one additional sentence or so ?

-Page 2 , line 16: "at calculating" -> "calculate"

-Page 2, line26: please remove Bechtold et al., 2001 and replace by Bechtold et al. 2008

-Page 2, line 34: "closure assumptions."->"closure assumptions and the entrainment of environmental air."

-Page 3, line 6: do not make paragraph

-Page 3, line 6: "out is"->"out"

-Page 3, line9:"parameterisation"->"parameterisations"

-Page 3, line 6: "mode, by"->"mode by"

-Page 3, line 11: "as regards"->"in" or "regarding"

-Page 3, line 15: "Tiedke"-."Tiedtke"

-Page 3, line 15: "estimate"->"estimates"

-Page 3, lines15-18: Please add a reference here on the usefulness on multi model or physics formulations

-page 4, line 19 and ff; "conseguence"->"consequence"

-page 4, line 24: 2forecast"->"forecasts"

-page 4, line 27: delete "one"

-page 4, line 30: rewrite as "...  for the 15 mm/6-h threshold, while for the highest thresholds (25, 50 mm/6-h), .."

-page 6, line 6: "the the"->"the"

-page 6, line 9: "seems to be .... ", rewrite as ".. are "mixed" appears to be appropriate to retain the advantages in both the lower and higher precipitation classes and/or shorter and longer forecast ranges."

-page 6, line 10: delete "It is worth pointing out" and start with "The implementation .."

-page 6, line 14: suggested rewrite is "identical, as should be its climatology,in order not to introduce systematic errors in the ensemble distribution"

---

## Referee Comment (RC2) · Anonymous Referee #2 · 16 Apr 2018

I regret I cannot recommend acceptance of the paper. That it is because its content is too elementary for an international journal, and because there is a major misconception from the authors' part concerning the interpretation of some of their results.

The paper presents an assessment of the compared performance of ensemble predictions for precipitation performed with the mesoscale limited-area COSMO (COnsortium for Small-scaleMOdelling) model, using two different convection schemes. The first one, developed by Tiedtke, is classical and has been operational in the COSMO model for a number of years. The other one has been developed more recently by Bechtold at the European Centre for Medium-Range Weather Forecasts (ECMWF).

The performance of the two schemes is, as said by the authors (p. 6, l. 10), *roughly indistinguishable*, with perhaps a slight advantage to the Tiedtke scheme. This result is not by itself of sufficient interest for publication in *Nonlinear Processes in Geophysics*. In addition, the study presented in the paper is too superficial for an international journal. The comparison of the two schemes in ensemble prediction of precipitation should have been preceded by a description of the precipitation regime produced by those two schemes. For instance, does the distribution of precipitation in the two schemes differ from each other, and from the observed precipitation ? Simple histograms of the precipitation amount, in the two schemes and in the observations, would in that respect be very instructive, whether the histograms mutually agree of differ. I presume that type of study has been performed (I do not personally know well enough the literature on convection parameterisation), and that part of the paper could consist mostly of appropriate references, with description of the main conclusions of the referenced papers. But appropriate explanations, or at least an appropriate discussion, must be presented as to the links between the properties of the two convection schemes and the performance of the ensemble predictions.

In addition, there is a major misconception from the authors' part concerning the interpretation of their diagnostics of outliers (Figures 6 and 9 and corresponding text). The authors seem to consider that the presence of outliers is undesirable (*A better performance of Cleps-10T, which has a lower number of outliers than Cleps-10B, can be noticed*, … p. 5, ll. 26-27). Well, it would be absurd to assume that a 10- or 20-ensemble defines bounds between which the verifying observation must necessarily lies. Statistically, there must be outliers, and the question of their number has been abundantly discussed in the literature. The usual assumption made in the validation of ensemble prediction systems in that the predicted ensemble is a sample of independent realizations of a probability distribution of which the verifying observation must be an additional, independent realization. That property is called *reliability*. If it is verified, the verifying observation must be statistically indistinguishable from the $N$ elements of the predicted ensemble. This means (among other things) that the probability for the observation to fall in any of the $N+1$ intervals defined by ranking the predicted ensemble values in increasing order must be the same for all intervals, and equal to $1/(N+1)$. The fraction of outliers must then be equal to $2/(N+1)$. That is 0.18 for $N = 10$, and 0.095 for $N = 20$. The fractions shown on the left panels of Figures 6 and 9 are smaller than that, especially at longer forecasts ranges. That means that the spread of the predicted ensembles are for both schemes significantly larger than the real uncertainty in the forecast. **That is in my mind a much more important conclusion than the minor differences observed between the two convection schemes**.

The right panels of figures 6 and 9 show the fraction of outliers lying above and below the upper and lower bounds of the ensembles. According to the same argument as above, that proportion must be equal to $1/(N+1)$, *i.e.* 0.09 and 0.048 for $N = 10$ and 20 respectively. Again, the fractions shown on the right panels of the figures are smaller than that, except for the 'max' curves of fig. 9, for which the observed fraction is correct.

The general conclusion is that both ensemble prediction schemes can mislead the user into expecting that the probability of occurrence of 'extreme' precipitation (either small or heavy) is larger than it really is, especially at longer forecast ranges (with as only exception for high precipitation in the right panel of fig. 9). Again, that is to me the main conclusion that must be drawn from the (limited amount of) results presented in the paper.

It may be of course that the authors do not consider that reliability of ensemble prediction systems (*i.e.*, overall statistical consistency between predicted probabilities of occurrence and observed frequencies of occurrence) is really important. But then they must explain clearly what makes in their minds that an ensemble prediction system is 'good', and why they consider that there must not be outliers.

I add two remarks.

- The right panels of Fig. 6 and 9 show a dissymmetry between low and high values of precipitation, without lower number of outliers below the minimum bound of the ensembles. This means that the ensembles, in addition to being overdispersed, are slightly biased towards low values of precipitation. These features may be visible (but not necessarily) on the global histograms I was mentioning above.
- The question of outliers, and more generally of the position of verifying observations with respect to predicted ensemble elements, has long been discussed, in particular through *rank histograms*. A rank histogram is, precisely, the histogram of the positions of verifying observations with respects with ensemble elements. The rank histogram of a reliable system must be flat. For a basic reference on rank histograms, see Hamill (2001).

**Reference**

Hamill, T. M., 2001: Interpretation of rank histograms for verifying ensemble forecasts. *Mon. Wea. Rev.*, **129**, 550-560

---

## Referee Comment (RC3) · Anonymous Referee #3 · 17 Apr 2018

Review of NPG-2018-21 : « Sensitivity of forecast skill to the parameterisation of moist convection in a limited-area ensemble forecast system »

General comments

The paper presents the impact of the use of two convection schemes (to represent uncertainties in the way convection is parameterized in numerical weather prediction models) in the COSMO system.
The manuscript is clear and well written but I think that some important points need to be clarified and reviewed before the paper can be accepted for publication.

Specific comments

When using a multi-physics approach to represent model error I think it is important to assess first each package of parametrization from a 'deterministic' point of view (using classical deterministic scores such as root mean square error, bias, heidke skill score, ...).
I suggest to perform two experiments (using unperturbed initial conditions), one using the Tiedke scheme, the other one using the Bechtold scheme, to compute some scores and to have a look at the distribution of precipitation. I think that the probabilistic evaluation that the authors use can not give informations about the behaviour of each of the schemes.

My main remark concerns the way 'outliers diagnostic' is computed and interpreted.
The 'outliers diagnostic' comes from the rank diagram score and represents the fraction of observations that lie outside the range of the ensemble.

My first comment concerns the way the rank diagram is constructed for precipitation forecast.
In an ensemble forecast of precipitation there are numerous cases in which one or several members forecast the same value. There also can be some cases in which all the members forecast a zero precipitation value.
How do the authors treat those cases ?
How do the authors treat the case in which the observed value and the minimum forecast value are equal ?
Do they add perturbations to the forecast values (as it is classically done in the literature) when one or several members forecast the same value ?

My second comment concerns the way the percentage of outliers is interpreted by the authors.
For a perfectly reliable ensemble of N members, the fraction of observations lying outside the range of the predicted values is 2/(N+1) (flat rank histogram).
Using a 10-member ensemble that is perfectly reliable, the fraction of outliers is 0.18 (0.095 for a 20-member ensemble).
Looking at figures 6 and 9 it can be seen that the percentage of outliers is below those theoritical values. In their comments of figures 6 and 9 the authors seem to consider that the lower the percentage of outliers, the better. This is wrong, especially if, as shown by the figures, the percentage of outliers is below the perfect theoritical value.

Looking at figures 6 and 9 we can only presume that the ensembles are over-dispersive (but this need to be confirmed by the rank histograms).

I think that using the percentage of outliers alone is not enough to properly evaluate the reliability of a forecast ensemble.

The authors should review all their comments of figures 6 and 9 and add, at least for one or two forecast ranges, rank histograms and comment their shapes (they can also use another score that measures the reliability of an ensemble).